

# Intra-subject sample size effects in plantar pressure analyses

Juliet McClymont[1], Russell Savage[1,†], Todd C. Pataky[2], Robin Crompton[1], James Charles[1] and Karl T. Bates[1]

[1] Department of Musculoskeletal & Ageing Science, Institute of Life Course & Medical Sciences, University of Liverpool, Liverpool, United Kingdom
[2] Department of Human Health Sciences, Kyoto University Graduate School of Medicine, Kyoto, Japan
† Deceased.

Corresponding author
Karl T. Bates,
k.t.bates@liverpool.ac.uk

## ABSTRACT

**Background**. Recent work using large datasets (>500 records per subject) has demonstrated seemingly high levels of step-to-step variation in peak plantar pressure within human individuals during walking. One intuitive consequence of this variation is that smaller sample sizes (e.g., 10 steps per subject) may be quantitatively and qualitatively inaccurate and fail to capture the variance in plantar pressure of individuals seen in larger data sets. However, this remains quantitatively unexplored reflecting a lack of detailed investigation of intra-subject sample size effects in plantar pressure analysis.

**Methods**. Here we explore the sensitivity of various plantar pressure metrics to intra-subject sample size (number of steps per subject) using a random subsampling analysis. We randomly and incrementally subsample large data sets (>500 steps per subject) to compare variability in three metric types at sample sizes of 5–400 records: (1) overall whole-record mean and maximum pressure; (2) single-pixel values from five locations across the foot; and (3) the sum of pixel-level variability (measured by mean square error, MSE) from the whole plantar surface.

**Results**. Our results indicate that the central tendency of whole-record mean and maximum pressure within and across subjects show only minor sensitivity to sample size >200 steps. However, <200 steps, and particularly <50 steps, the range of overall mean and maximum pressure values yielded by our subsampling analysis increased considerably resulting in potential qualitative error in analyses of pressure changes with speed within-subjects and in comparisons of relative pressure magnitudes across subjects at a given speed. Our analysis revealed considerable variability in the absolute and relative response of the single pixel centroids of five regions to random subsampling. As the number of steps analysed decreased, the absolute value ranges were highest in the areas of highest pressure (medial forefoot and hallux), while the largest relative changes were seen in areas of lower pressure (the midfoot). Our pixel-level measure of variability by MSE across the whole-foot was highly sensitive to our manipulation of sample size, such that the range in MSE was exponentially larger in smaller subsamples. Random subsampling showed that the range in pixel-level MSE only came within 5% of the overall sample size in subsamples of >400 steps. The range in pixel-level MSE at low subsamples (<50) was 25–75% higher than that of the full datasets of >500 pressure records per subject. Overall, therefore, we demonstrate a high probability that the very small sample sizes ($n < 20$ records), which are routinely used in human and animal studies, capture a relatively low proportion of variance evident in larger plantar pressure data set, and thus may not accurately reflect the true population mean.

## INTRODUCTION

The difficulty of directly assessing internal motion and forces in the distal limbs of humans and animals without resorting to invasive approaches (*Lundgren et al., 2008*), means that external measures of foot mechanics, such as pressure records, are currently crucial to our understanding of foot function (e.g., *Frykberg et al., 1998*; *Pataky et al., 2008*; *D'Août et al., 2009*; *Crompton et al., 2012*; *Bates et al., 2013a*; *DeSilva & Gill, 2013*). Pressure records are used to diagnose and evaluate abnormalities in the feet and lower limbs of humans (e.g., *Frykberg et al., 1998*; *Pham et al., 2000*; *Boulton, Kirsner & Vileikyte, 2004*) and companion animals (e.g., *Stadig, Lascelles & Bergh, 2016*; *Romans et al., 2004*), to identify therapeutic interventions (e.g., *Arts & Bus, 2011*; *Bus et al., 2008*; *Paton et al., 2011*; *Melia et al. 2021*) and in furthering our understanding of fall avoidance in the elderly (*Xi et al., 2020*). They provide key insights into the modulation of foot function (e.g., *Simpson et al., 1993*; *Maluf & Mueller, 2003*; *Pataky et al., 2008*; Stepháne 2008; *Caravaggi, Leardini & Giacomozzi, 2016*; *Taş & Çetin, 2019*), in addition to delivering fundamental insights into the evolution of hominid foot morphology and function (e.g., *Vereecke et al., 2003*; *Crompton et al., 2012*; *Bates et al., 2013a*; *Bates et al., 2013b*; *DeSilva & Gill, 2013*; *McClymont & Crompton, 2021*). Recent work using large intra-subject human datasets (>500 steps per subject) identified high levels of both inter- (i.e., between individuals) and intra-subject (i.e., within subjects step-to-step) variance in peak plantar pressure in the midfoot (*Bates et al., 2013a*), and across the whole plantar surface in healthy adults (*McClymont et al., 2016*). This is qualitatively consistent with earlier work measuring variability in loading patterns of neuropathic patients from sample sizes of >50 pressure records (*Cavanagh et al., 1998*). Qualitatively, these findings would imply that a large number of steps might be required to represent the variability and central tendency of an individual's plantar pressure characteristics with a high degree of accuracy.

Previous studies that assess intra-subject variability in gait kinematics suggest multiple steps should be analysed to with the aim of capturing the variability of a participant's gait patterns (*Cavanagh et al., 1998*; *Dingwell et al., 2001*; *Owings & Grabiner, 2003*; *Hausdorff, 2007*; *Jordan, Challis & Newell, 2007*; *Bruijn et al., 2013*; *Riva, Bisi & Stagni, 2014*). For example, *Owings & Grabiner (2003)* established 400 steps per subject as a necessary minimum to reliably characterise the motor patterns determining stride length and stride width of an individual participant. While kinematic and spatiotemporal gait parameters have been extensively studied, there has been relatively little research on intra-subject sample size effects in analyses of plantar pressure. The 'one-step', 'two-step', and 'three-step' protocols (*Van der Leeden et al., 2004*) and the 'five-step mid-gait' approach (*McPoil et al., 1999*) are comparable as protocols (*Oladeji et al., 2008*) but are designed to limit small intra-subject sample sizes to characteristically $n$ <10 steps per subject (e.g., *Taylor, Menz & Keenan, 2004*). Such protocols are necessary for patients with a low capacity for plantar loading (e.g., patients with diabetic neuropathy or plantar ulcers), but they are

still employed in research testing protocols for healthy adults (*Taylor, Menz & Keenan, 2004*; *Bus & de Lange, 2005*; *Phethean et al., 2014*; *McKay et al., 2017*; *Sole et al., 2017*), and children (*Hennig & Rosenbaum, 1991*; *Bosch, Gerss & Rosenbaum, 2010*; *Phethean & Nester, 2012*). *Arts & Bus (2011)* conducted a clinical study on patients with diabetic neuropathy assessing variability in numerous plantar pressure variables using an in-shoe Pedar system, orthopedic shoes and non-consecutive steps. Collecting a total sample of 20 non-consecutive steps per foot, the authors concluded that 12 mid-gait steps per foot were sufficient for reliably capturing in-shoe peak pressure patterns (*Arts & Bus, 2011*). *Kernozek, LaMott & Dancisak (1996)* suggested that 8 steps per foot were required for reliable in-shoe plantar pressures in healthy participants during steady-state treadmill walking. Kiejers et al., (2009) collected 17 steps per subject (using young, healthy individuals and the three-step protocol) and suggested that, on average, 3.8 steps per subject were required to meet acceptable levels of reliability in their study context. Where larger sample sizes ($n = 200$ steps) have been considered, only one pressure metric (maximum peak pressure) was evaluated for sensitivity to the number of steps used to quantify the central tendency of individual participants (*Melvin et al., 2014*). Concerns about sample size have been highlighted by *McClymont & Crompton (2021)* in the study of fossil footprints (a parallel field to pressure analysis; e.g., *Crompton et al., 2012*; *Bates et al., 2013b*), in particular in cases where locomotor behaviour and function are interpreted from a very small sample size of sequential fossil footprints (e.g., *Hatala et al., 2016*), as usually is the case in this field. Therefore, it is currently unknown how the observed variance in different plantar pressure metrics reported from small sample sizes of steps ($n < 50$), compares to that reported from large sample sizes ($n > 500$) using different pressure metrics in barefoot treadmill walking.

That intra-subject sample size effects have not been extensively studied despite this breadth of application is perhaps not surprising given that many experiments pose unique and inherent challenges that may restrict sample size, and influence the characterisation of plantar pressure. For example, in non-companion animal research, intra-subject sample size is limited by access to animals and their willingness to partake in, and follow experimental protocols, limitations that are also commonplace in human paediatric research (*Price et al., 2018*). In clinical and veterinary contexts, the number of steps per subject is often restricted by physical limitations and efficacy levels that are unique to each patient. As with all kinetic and kinematic gait parameters, valid interpretations of pressure records rests on the assumption that either a single step, or sample of steps, reliably characterises the foot mechanics of the individual, or population under study, to a degree satisfactory for each analysis. A clear picture of the variance of each subject should be delineated for robust interpretations to be made.

The aim of this study is to explore the sensitivity of a number of plantar pressure metrics to intra-subject sample size (number of steps per subject) using a random subsampling analysis. Our goal is to generate insight into sample size-related variability and thereby provide guidance to the design of future plantar pressure analyses.
## MATERIALS & METHODS

### Data collection

We used an existing, freely available human plantar pressure dataset (*McClymont et al., 2016*; doi: http://doi.org/10.5061/dryad.09q2b), because of the size and range of walking speeds it encompassed (i.e., >475 pressure records per subject at 1.1m/s and, >500 pressure records per subject per speed at 1.3−1.9 m/s). According to *McClymont et al. (2016)* sixteen healthy participants (aged 21–46) with no known or prior limb abnormalities walked barefoot continuously for five minutes, at five different speeds (1.1m/s, 1.3 m/s, 1.5m/s, 1.7m/s and 1.9 m/s) in random order, on a Zebris FDM-THM pressure sensing treadmill. Anthropometric properties of the dataset are described in *McClymont et al. (2016)*.

### Data processing

Within-footfall maximum pressure values were extracted to yield between 2780–3535 pedobarographic records (pressure records; 1 per step) per subject in total from the five speed trials. Participants with shorter leg lengths took more steps in 5 min, producing more pressure records per speed trial, than those with longer leg lengths (Tables S1–S2). For each subject, all pressure records from each speed trial were extracted, stacked and registered to each other, using an algorithm that minimises the mean squared error (MSE) between individual pressure records in the stack so that homologous structures optimally overlapped (*Pataky & Goulermas, 2008*). Pressure records that varied in length by more than 1.5x standard deviations of the mean length or width from each trial were automatically removed from the data set during the registration process to remove anomalous footfalls and footfalls where a portion of the subject's footfall overlapped the edge of the pressure plate embedded in the treadmill. The five registered datasets per subject could then be used to explore the impact of the intra-subject sample size (the number of steps per subject) on foot pressure metrics using our custom written, pixel-wise software, Pedobarographic Statistical Parametric Mapping (pSPM) (*Pataky & Goulermas, 2008*; *Pataky et al., 2008*), in MATLAB (MathWorks, USA).

### Data analysis

Investigation of the relationship between sample size and intra-subject plantar pressure variance first required quantitative pressure metrics, and subsequently an analysis of how each metric varies according to the number of steps included in the analysis. Broadly speaking, there are two branches of analyses available to quantify plantar pressure characteristics; simple metrics, where a small number of global (e.g., the maximum and mean pressure from each steps) or regional metrics (e.g., the pressure from the most central pixel in the 'heel' region, the average pressure from pixels in the 'hallux' region); and, pixel-level, vector analyses (e.g., pSPM; *Pataky & Goulermas, 2008*; *Pataky et al., 2008*) that attempt to quantify pressure characteristics from the whole plantar surface of the foot without spatial data reduction. Here we attempted to investigate the sensitivity of both these types of approaches to intra-subject sample size (Fig. 1). For simple, singular pressure metrics we carried out two intra-subject sample size analyses based on (1) the mean
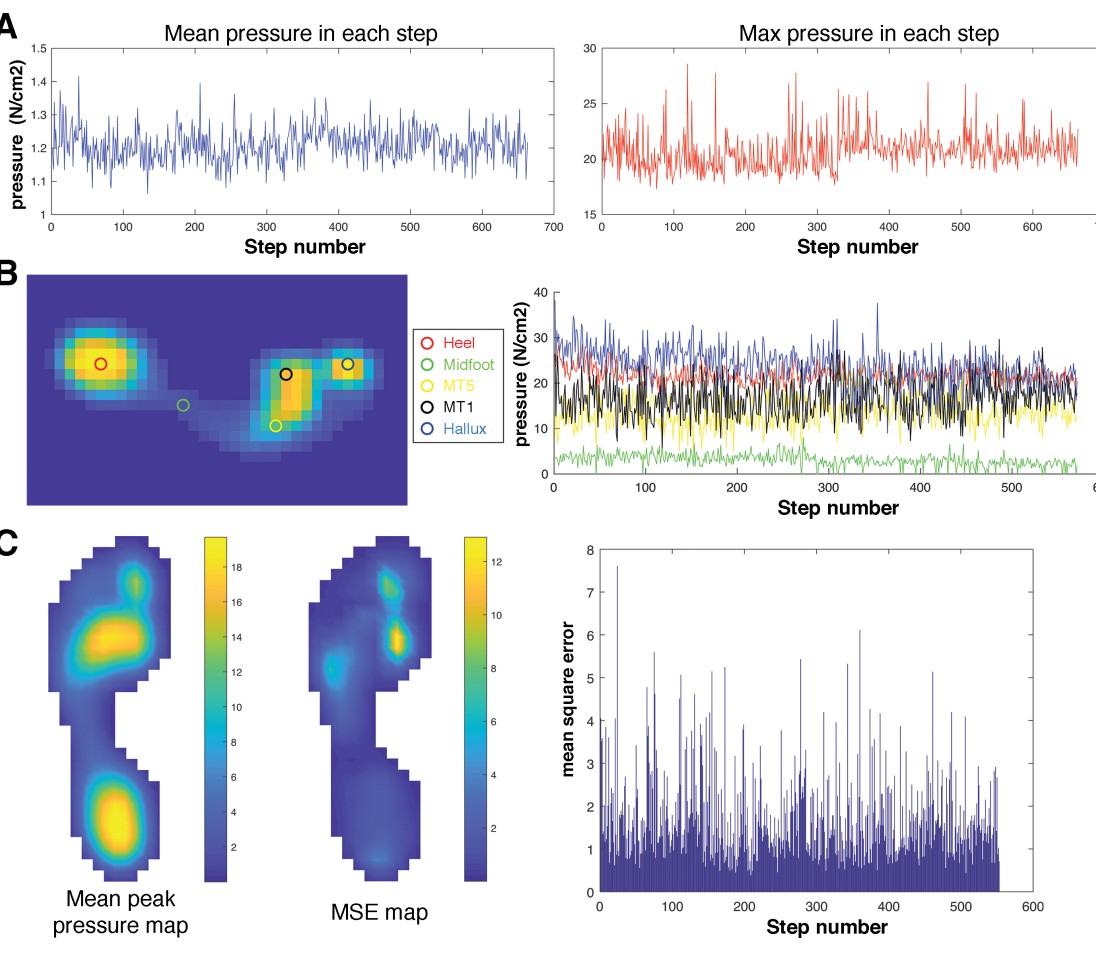

**Figure 1  Overview of the plantar pressure metrics used in the intra-subject (number of steps) subsampling analysis.** We investigated the sensitivity of three different metric types to the number of steps analysed within each subject. These were (A) the simple 'whole-foot' mean (mMEANP) and maximum pressure (mMAXP), and (B) the absolute values of pixels representing the approximate centroid of the heel, midfoot, lateral and medial forefoot (MT5 and MT1 above) and the hallux. (C) To examine how the number of steps analysed within a subject may influence the foot as a whole (i.e., the sum of variability across all pixels in each record), and therefore sample size implications for pixel-level vector analyses of pressure, we calculated the global mean square error (MSE) of all pixels in each pressure record following *Mc-Clymont et al. (2016)*.

and maximum pressure in each peak plantar pressure within each subject at each speed (Fig. 1A), and (2) the pressure of a single pixel that approximates the centre of 5 locations found commonly in regional analyses ('heel', 'midfoot', 'distal head of the first metatarsal' [MT1], 'distal head of the fifth metatarsal' [MT5], and 'hallux') (Fig. 1B). To examine how the number of steps analysed within a subject may influence whole-foot pressure characteristics, and therefore sample size implications for pixel-level vector analyses of pressure (e.g., *Pataky & Goulermas, 2008*; *Pataky et al., 2008*; *Bates et al., 2013a*; *Bates et al., 2013b*), we calculated the global mean square error (MSE) of all pixels in each pressure record following *McClymont et al. (2016)* (Fig. 1C). Specifically, the MSE was calculated

over every non-zero pixel in each pressure record within the overall total sample from all five speeds according to Eq. (1):

$$\mathrm{MSE} = \frac{1}{N} \sum_{k}^{N} (I_{0k} - I_{1k})^2 \tag{1}$$

where $N$ is the total number of non-zero pixels in the mean image; $I_0$ is the mean of the subject's overall sample and $I_1$ is an individual pressure record. The MSE of each pixel was then summed to produce a total MSE value for each individual pressure record about the subject's overall mean pressure record (Fig. 1C).

To explore the sensitivity of these metrics to the number of steps per subject, we conducted a random subsampling analysis (Fig. 2). 1000 random subsamples of pressure records were extracted from each subject's overall total speed trial at subsamples of 5, 10, 25, 50, 100, 200, 300, 400 and 500 steps (i.e., 1000 randomly generated samples with an $n$ steps of 5, 1,000 randomly generated samples with an $n$ steps of 25, 1000 randomly generated samples with an $n$ steps of 50 etc.). The range in mean maximum pressure (mMAXP), the mean of the mean pressures (mMEANP) and mean MSE from within each 1,000 subsample at each $n$ steps then was then calculated with respect to the subject's overall mean pressure record (taken from the total sample size of >500 steps), enabling quantitative comparison of the sensitivity of each metric to the manipulation of intra-subject sample size (Fig. 2). We ran these subsampling analyses from each speed defined in the dataset (1.1 ms$^1$, 1.3 ms$^1$, 1.5 ms$^1$, 1.7 ms$^1$ and 1.9 ms$^1$) to test for possible changes in response to intra-subject sample size at different walking speeds. Because results remained consistent across all these speeds, we focus on on the results at 1.3 m/s subsequent sections, as it represents a walking speed easily and regularly practiced by healthy young adults in everyday life (*Himann et al., 1988*; *Oberg, Karsznia & Oberg, 1993*; *Samson et al., 2001*). The results from speeds 1.1 m/s, 1.5 m/s, 1.7 m/s and 1.9 m/s, presented separately in the supplementary material (Figs. S1–S8).

## RESULTS

### Sensitivity of simple 'whole-foot' pressure metrics to intra-subject sample size

Across all speeds, all participants showed a qualitatively similar pattern of response to random subsampling of mMEANP and mMAXP (Fig. 3A). The average range (for the 16 subjects) in the mMAXP values from 1000 randomly subsampled populations at 1.3 m/s, remained less than 2N/cm$^2$ for sample sizes of 100–500 steps per subject but increased sharply when the number of steps for an individual subject was reduced to 50 or less (Fig. 3A). At 5 steps per subject, the average range in mMAXP values was 7.1 N/cm$^2$, with the highest and lowest subjects yielding ranges of 4.7 N/cm$^2$ and 10.3 N/cm$^2$ (Fig. 1A). These values at an $n$ of five steps corresponded to percentage errors of 13.2% for the average across the subjects, and 8.5% and 18.2% for the highest and lowest subjects (Fig. 3C). Similar trends were seen in the range in mMEANP (Figs. 1B & 1D); the average range across subjects remained below 0.1N/cm$^2$ for sample sizes of 100-500 steps per subject, but

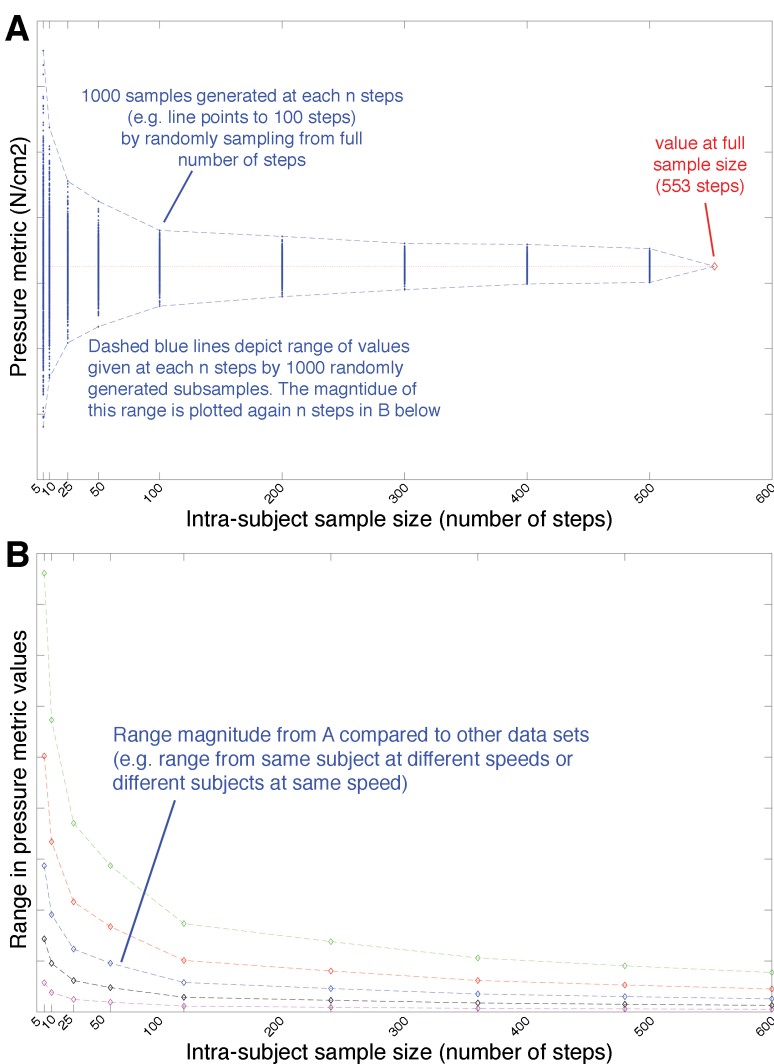

**Figure 2** **Conceptual example of the subsampling analysis applied to the various pressure metrics shown in Fig. 1.** (A) 1000 random subsamples of the pressure metrics were extracted from each subject's overall total number of steps at subsamples of 5, 10, 25, 50, 100, 200, 300, 400 and 500 steps (represented as blue diamond data points in A). As the number of steps included in any one of 1000 subsampled populations decreases it would be expected that the range in the absolute value of the metric pressure metric would increase, reflecting (for example) the relatively greater impact of 'extreme' high and low values from specific steps. In other words, the absolute range of values for the pressure metric (dashed blue lines delimiting the maximum –minimum values from the 1000 subsampled populations) would be expected to increase as the number of steps included decreases, as shown in (B). The absolute and relative magnitude of metric ranges are plotted against sample size (n steps) in subsequent plots to make comparisons across subjects and speeds, as shown in (B).

increased sharply when the number of steps was reduced to 50 or less (Fig. 3B). At 5 steps per subject, the average range in mMEANP values exceeded 0.3 N/cm$^2$, with the highest and lowest subjects yielding values of 0.1 N/cm$^2$ and 0.5 N/cm$^2$ (Fig. 3B). A corresponding percentage error of 7.7% was found for the average across the subjects, with 4.8% and 10.4% for the highest and lowest subjects (Fig. 3D).

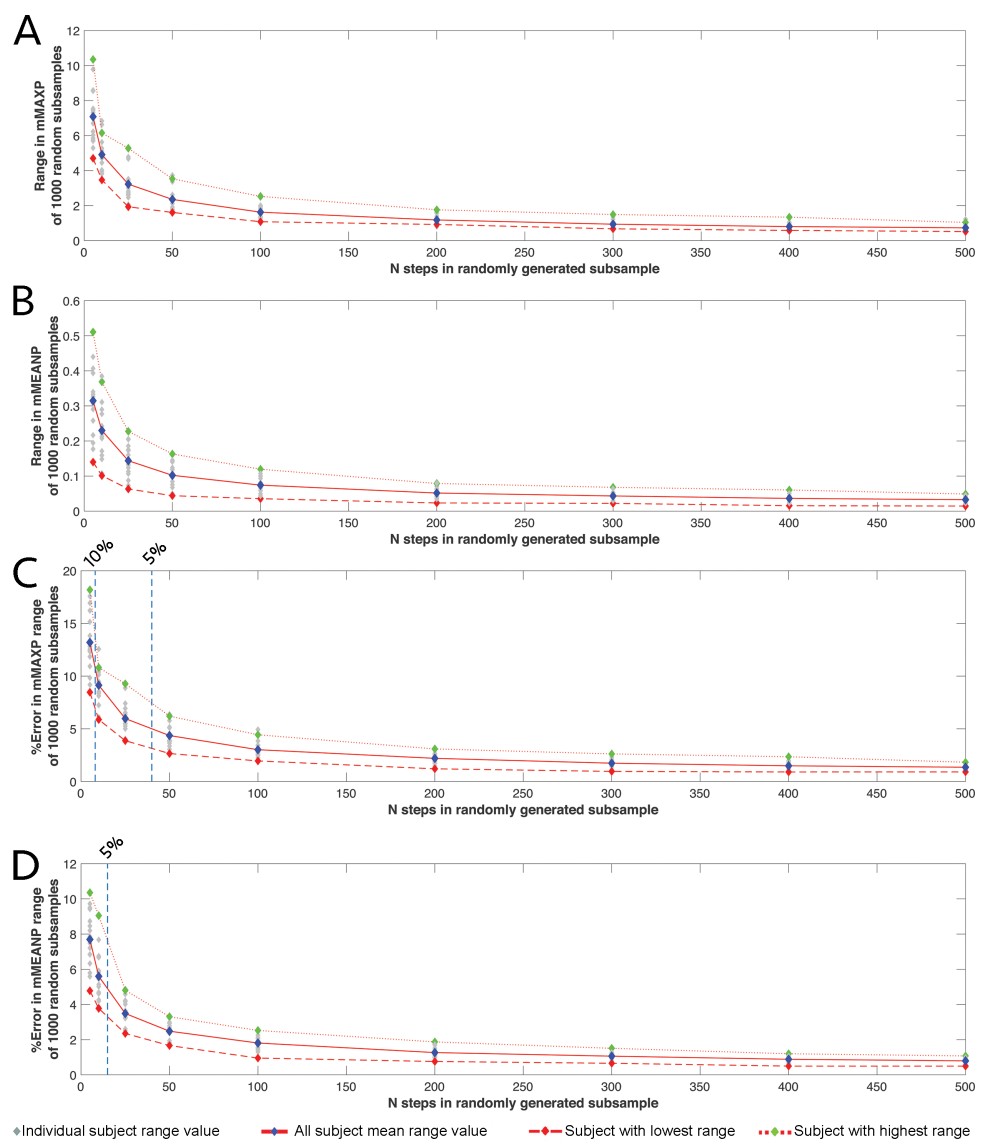

**Figure 3** **The relationship between intra-subject sample size (number of steps per subject) and the range of values for simple pressure metrics.** The relationship between intra-subject sample size (number of steps per subject) and the range in (A) absolute mMAXP, (B) absolute mMEANP, (C) relative mMAXP and (D) relative mMEANP at each subsample given by 1000 randomly generated subsamples at a walking speed of 1.3 m/s. Data for individual subjects are shown as grey diamonds, with the average for all subjects (blue diamonds and red solid line) and the subjects with the highest (green diamonds and thin dashed red line) and lowest (red diamonds and thick dashed red line) ranges highlighted. Dashed vertical lines on C and D indicate the *n* steps at which percentage values of the data mean reach specific thresholds (e.g., 5%, 10%) of the value at the full sample size.

To place these magnitudes in a comparative context, Figs. 4A & 4B show how the range of mMEANP and mMAXP values given by the 1,000 randomly generated subsamples, changes across 5–500 steps within a single subject at all five walking speeds. Only at sample sizes of 200 or more steps were the qualitative difference between speeds (i.e., higher mMEANP

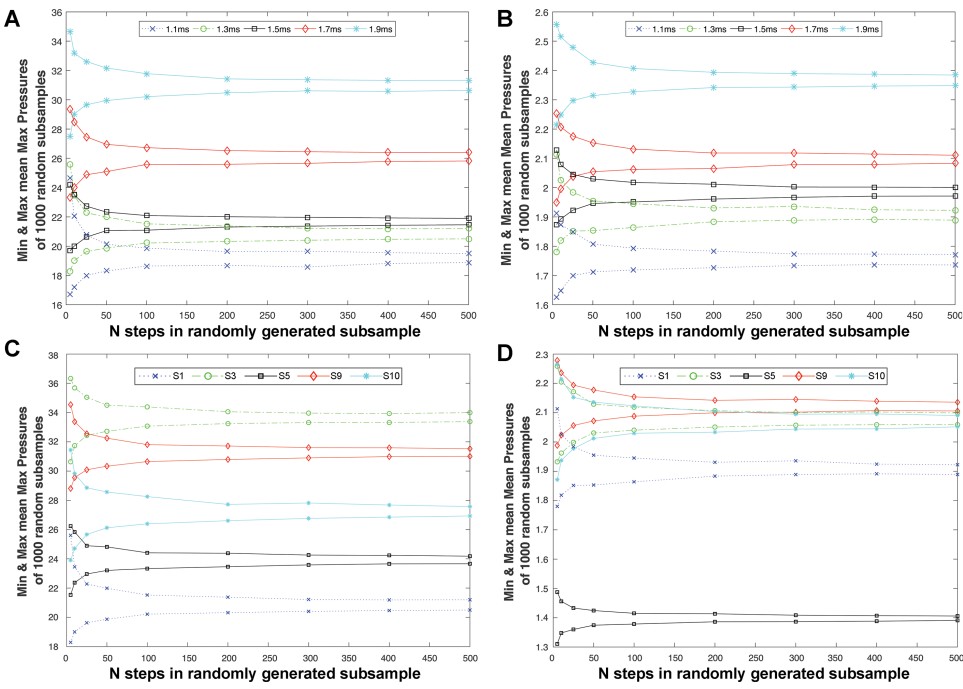

**Figure 4** **Intra-subject sample size effects on the patterns recovered when assessing changes in pressure with speed within subjects, and in relative pressures across subjects.** The relationship between intra-subject sample size (number of steps per subject) and (A) the minimum and maximum values recovered for mMAXP and (B) mMEANP in Subject 1 across a range of walking speeds ($1.1-1.9$ m/s). These plots demonstrate that only at sample sizes of 200 or more steps were the qualitative difference between speeds (i.e., higher mMEANP and mMAXP values at higher walking speeds) preserved. At 200 or fewer steps for mMAXP and 50 or fewer steps for mMEANP, the range in values produced at adjacent walking speeds began to overlap, raising the possibility of comparative analyses failing to correctly identify the qualitative difference between walking speeds in this subject if such sample sizes were used. Plots of (C) mMAXP and (D) mMAXP for different subjects can be used to assess the predictive capability of different sample sizes to correctly recover inter-subject differences in pressure at a single speed (1.3 m/s). For (C) mMAXP, the correct qualitative differences between subjects were preserved until the number of steps was reduced to 25 or less, whereas for (D) mMEANP, the situation is more complex, but most subjects begin to show overlapping ranges at steps of 200 or less, with high levels of overlap (and therefore potential for incorrect identification of relative pressures) at 25 or fewer steps.

and mMAXP values at higher walking speeds) preserved. At 200 or fewer steps for mMAXP and 50 or fewer steps for mMEANP (Figs. 4A–4B), the range in values produced at adjacent walking speeds began to overlap, raising the possibility of comparative analyses failing to correctly identify the qualitative difference between walking speeds (i.e., falsely suggesting a decrease in mean and maximum pressure with increasing walking speed). Figs. 4C–4D show the range of mMEANP and mMAXP values given by the 1,000 randomly generated subsamples changes across 5-500 steps for five randomly selected subject walking at 1.3 m/s. For mMAXP, the correct qualitative differences between subjects were preserved until the number of steps was reduced to 25 or less, when the ranges between adjacent subjects overlap, indicating the potential for an analysis to incorrectly identify relative pressure differences between subjects (Fig. 4C). For mMEANP, the situation is more complex:

subject 5 shows very low mMEANP at 1.3 m/s and their range in pressure magnitudes remained lower than all other subjects regardless of the number of steps used (Fig. 4D). However, the other four randomly selected subjects began to show overlapping ranges in mMEANP at steps of 200 or less, with high levels of overlap (and therefore potential for incorrect identification of relative pressures) at 25 or fewer steps (Fig. 4D).

### Sensitivity of single-pixel region centroid values to intra-subject sample size

Across all speeds, all participants showed a qualitatively similar pattern of response to random subsampling of mean values for the centroid pixels of the five-foot regions: 'heel', 'midfoot', lateral ('MT5') and medial ('MT1') forefoot, and 'hallux' (Fig. 1B), although the absolute and relative magnitudes of responses varied considerably across regions and between subjects (Figs. 5–6). The average range (for the 16 subjects) in the mean values given by the 1000 randomly subsampled populations at 1.3 m/s remained less than $3N/cm^2$ for sample sizes of >200 steps per subject but, increased sharply in all five regions when the number of steps for an individual subject was reduced to 100 or less (Fig. 5). At five steps per subject, the average range in mean values exceeded $8N/cm^2$ in all five regions, with the highest absolute values occurring in the medial forefoot ('MT1') and hallux (Figs. 5D–5E) and the lowest absolute values in the 'midfoot' (Fig. 5B). Within each region there was considerable inter-subject variation in absolute range magnitudes versus subsample size. For example, at five steps per subject, the subjects with largest and smallest absolute ranges yielded values of $9.74 N/cm^2$ versus $0.97N/cm^2$ in the 'midfoot' (Fig. 5B), and $22.12 N/cm^2$ versus $5.24N/cm^2$ in the medial ('MT1') forefoot (Fig. 5D) respectively. When ranges were normalized as percentage of the pixel values at the full sample size, the 'midfoot' centroid pixel shows the highest relative error of all five regions (Fig. 6) with a mean value for all subjects of 171%, and 672% versus 41.74% for the subjects with highest and lowest values (Fig. 6B). The lowest percentage values were seen in the 'heel' centroid pixel, where the mean values for all subjects was 12.16%, with the highest and lowest subject values recovered as 22.4% versus 3.07% (Fig. 6A).

### Sensitivity of whole-foot MSE to intra-subject sample size

At 1.3 m/s, the range of whole-foot mean MSE given by the 1000 randomly generated subsamples reduced exponentially as subsampled number of steps increased (Fig. 7A, Figs. S1–S8, Tables S1–S2). All results for other speeds follow the same exponential curve (see Fig. S1–Fig. S8). The number of pressure records required for the range in mean MSE to be within 5% of the overall dataset mean MSE range was $n = 401$ records at 1.3 m/s (Fig. 7B, Tables S1–S2). In smaller subsamples (i.e., $n = <10$) the range in mean MSE exceeded 75% of the overall sample range in mean MSE (Fig. 7).

## DISCUSSION AND CONCLUSIONS

The aim of this study was to understand the stability of a number of widely used plantar pressure metrics (Fig. 1) when exposed to random manipulations of sample size (Figs. 3–7). To this end, we hoped to provide insight into the effect that intra-subject sample size (the
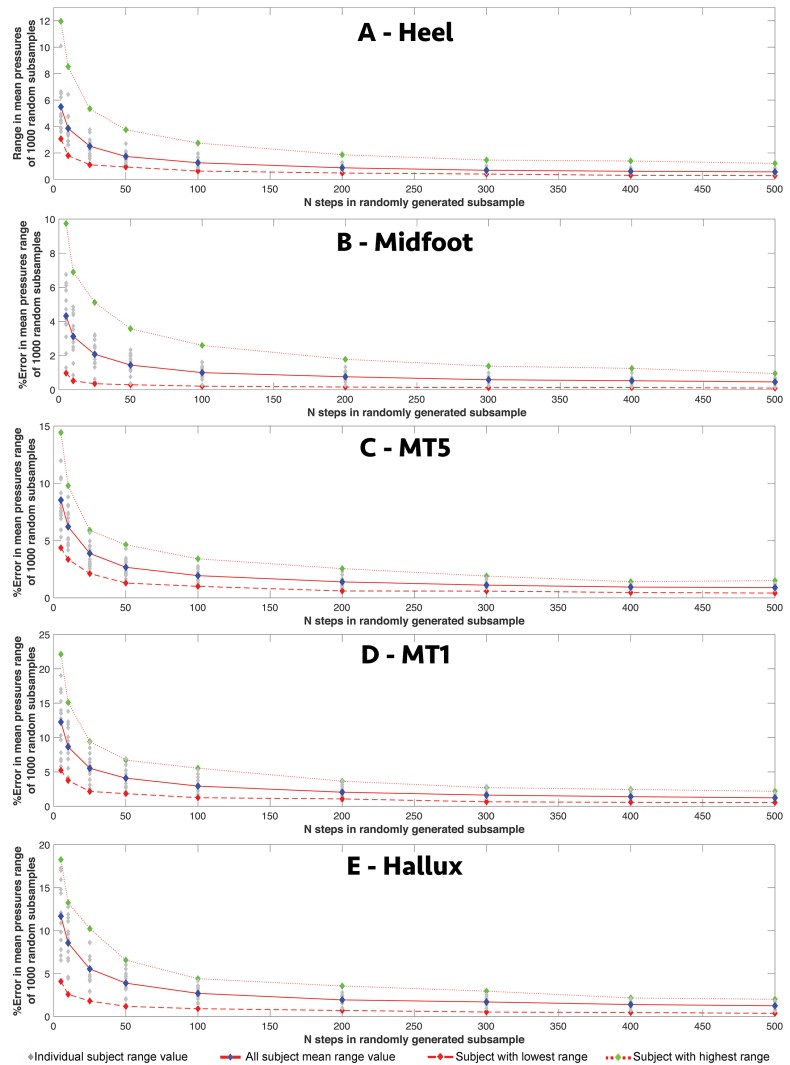

**Figure 5** **The relationship between intra-subject sample size (number of steps per subject) and the range in absolute pressure values for different foot regions.** The relationship between intra-subject sample size (number of steps per subject) and the range in absolute mean values for the centroid pixel in the (A) heel, (B) midfoot, (C) lateral (MT5) and (D) medial (MT1) forefoot, and (E) hallux in all subjects at 1.3 m/s.

number of steps analysed within a subject) has on the estimative and interpretive power in assessing variance in peak plantar pressure, to provide general guidance to the design of plantar pressure analyses.

The level of accuracy or reliability required for an analysis depends on the goals of the experiment or the specific hypotheses being tested. The effects of the number of steps analysed per subject derived herein provide general guidance in this respect for a number of metrics in healthy human participants during steady-state, barefoot, treadmill walking. During continuous speed trials, the observed variation in each pressure record is not entirely independent from each other, rather each step is part of a feedback loop in the

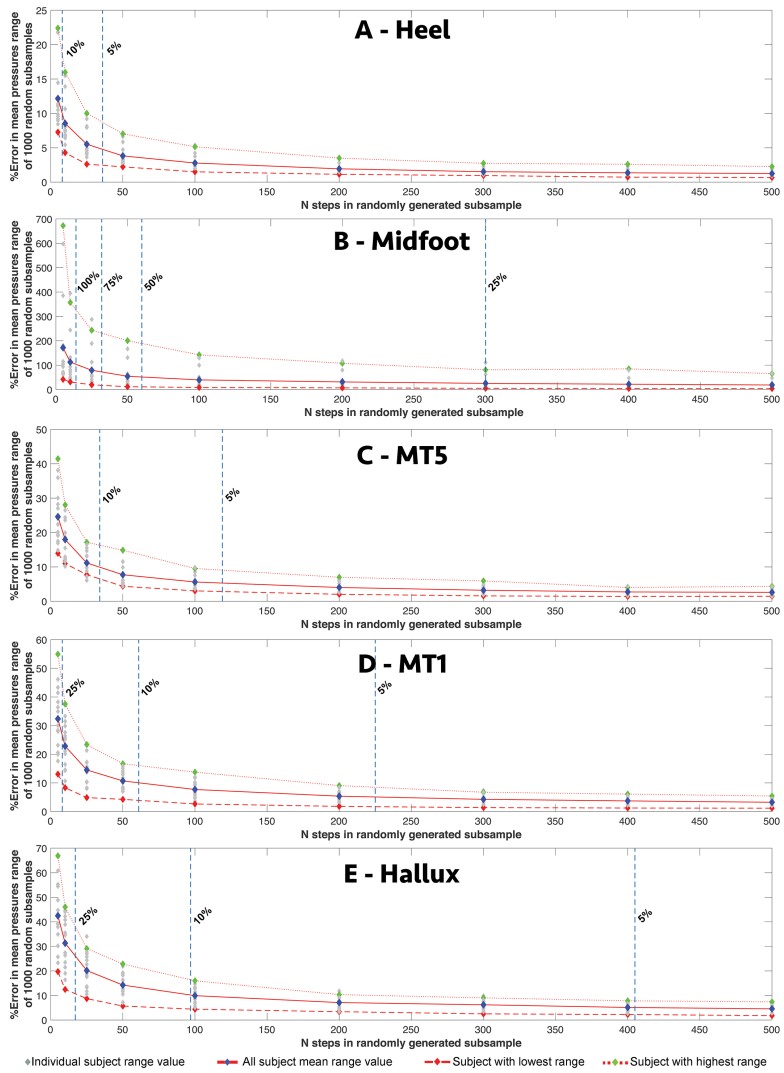

**Figure 6** **The relationship between intra-subject sample size (number of steps per subject) and the range in relative pressure values for different foot regions.** The relationship between intra-subject sample size (number of steps per subject) and the range in relative mean value for the centroid pixel in the (A) heel, (B) midfoot, (C) lateral (MT5) and (D) medial (MT1) forefoot, and (E) hallux in all subjects at 1.3 m/s.

motor control of walking, whereby a walking pattern in any gait cycle may influence subsequent strides in accordance with fractal dynamics (*Terrier & Olivier, 2011*). Thus, the continuous collection of large numbers of pressure records reduces the risk of observing higher variability than would be reported from smaller sample sizes. Higher variability will likely be observed from short recording sessions with a low intra-trial *n*. However an analysis of this effect from non-continuously collected pressure records remains to be performed.

In this study we chose to compare variability in a number of different pressure metrics (Fig. 1) across incrementally subsampled numbers of steps per subjects collected during five

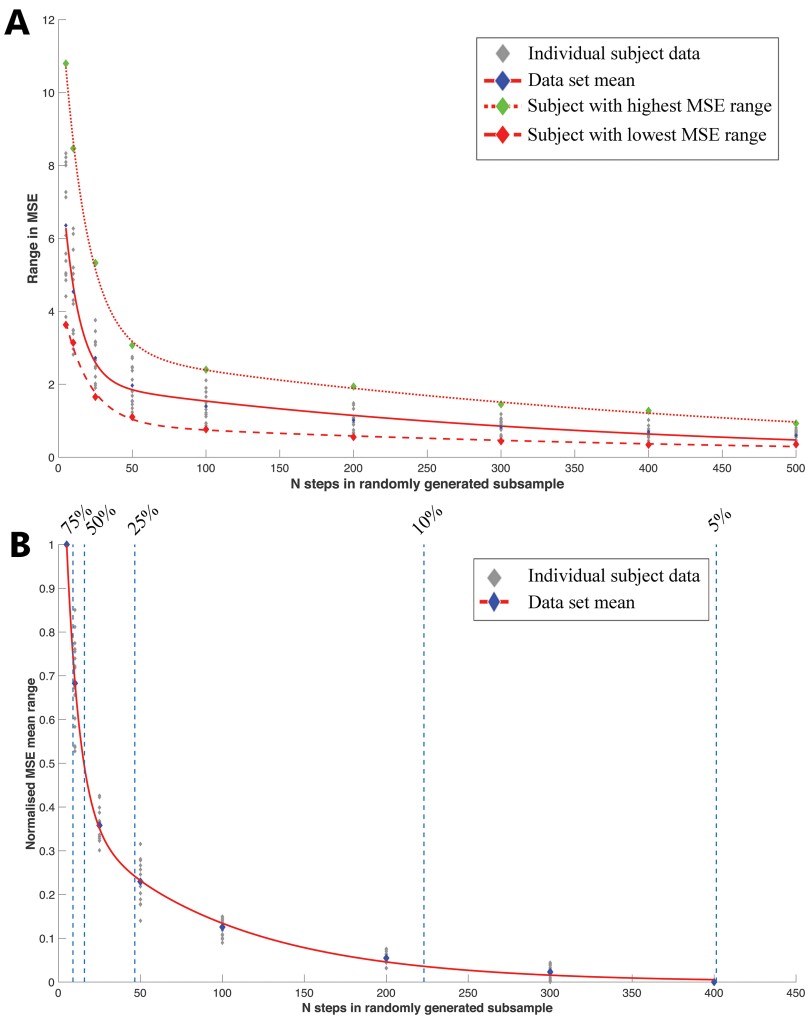

**Figure 7  The relationship between intra-subject sample size (number of steps per subject) and the range in 'whole-foot' plantar pressure mean MSE.** (A) The relationship between intra-subject sample size (number of steps per subject) and the range in mean MSE at each subsample given by 1000 randomly generated subsamples at a walking speed of 1.3 m/s. This relationship is well described by exponential curves in all subjects, which are plotted here for the overall mean of all subjects and the subjects with the highest (7) and lowest (13) overall MSE. (B) The relationship between intra-subject sample size (number of steps per subject) and the normalized range in mean MSE at each subsample. The normalized range in MSE is plotted as a percentage of mean MSE of the full dataset for each subject walking at 1.3 m/s. When the sample size is $n \Rightarrow 400$ steps, then the range in mean MSE is less than 5% higher than the full dataset values for each subject. The observed range in mean MSE at smaller dataset sizes ($n = 25$ steps) is more than 50% higher than the total dataset values observed for each subject.

minutes of continuous walking at a range of walking speeds (1.1−1.9 m/s). However, we recognise that our experiment does not reflect many other circumstances in biomechanics research where plantar pressure is analysed. In some human studies (e.g., *Kernozek, LaMott & Dancisak, 1996*; *Cavanagh et al., 1998*; *Frykberg et al., 1998*; *Pham et al., 2000*; *Boulton, Kirsner & Vileikyte, 2004*; *Bus et al., 2008*; *Paton et al., 2011*; *Arts & Bus, 2011*) and in animal research (*Romans et al., 2004*; *Michilsens et al., 2009*; *Stadig, Lascelles &*

*Bergh, 2016*; *Panagiotopoulou et al., 2016*), plantar pressure is typically recorded during unrestricted over-land locomotion using free-standing pressure mats or plates. In these circumstances the environmental conditions that the participant encounters inherently differs from treadmill locomotion, and furthermore non-sequential steps are common. Caution should therefore be taken in applying our treadmill results to overground datasets until further tests can confirm the effects. However, we would predict broadly similar exponential increases in variance with a small number of steps if our subsampling protocol were applied to such data sets.

Our analysis of simple 'whole-foot' pressure metrics (Figs. 1A, 3–4) demonstrates a complex interaction between the number of steps used per subject and the specific pressure metric chosen (mMEANP versus mMAXP) to represent a step with a single aggregate value. In general, our analysis suggests that at 200 steps or more, the absolute and relative 'error' in both mMEANP and mMAXP relative to a sample of 500–600 records is likely to be very low (Figs. 3–4). As a result, 200–500 records (in our analysis) yield the same qualitative patterns or results, both in the intra-subject analysis of pressure magnitude across walking speeds (Figs. 4A–4B), and, in a cross-subject comparison at 1.3 m/s (Figs. 4C–4D). At fewer than 200 steps (and particularly at <50 steps) the absolute and relative 'error' in simple pressure metrics relative to a sample of 500–600 records, begins to increase approximately exponentially (Figs. 3–4). mMAXP appears to yield greater error magnitudes than mMEANP (Figs. 3C–3D), yet in our comparative analysis, mMEANP was clearly more susceptible to qualitative error in terms of the potential to incorrectly represent relative differences in pressure across subjects at lower samples sizes (Fig. 4D).

The mean value of single pixels at the centroid of our five regions (Fig. 1B) vary considerably in their absolute (Fig. 5) and relative (Fig. 6) response to subsampling. Perhaps not surprisingly, higher pressure regions, particularly the medial midfoot ('MT1') and hallux, are recovered with the highest absolute ranges as increasingly fewer steps are used to derive the mean pressure (Figs. 5D–5E). This finding is consistent with the MSE 'variation maps' (Fig. 1C) for this same subject data set presented by *McClymont et al. (2016)*, which demonstrated that average MSE was broadly correlated with pressure magnitudes across the foot and therefore highest in the medial forefoot and hallux. *McClymont et al. (2016)* also examined the spatial variation in co-efficient of variation (CoV) of peak pressure across the whole plantar surface of the foot. These CoV variation maps are consistent our relative assessment of intra-subject sample size effects in different foot regions (Fig. 6), which is perhaps not surprising given that both are forms of normalised measurements. Normalised assessments suggest that step-to-step variability (*McClymont et al., 2016*) and subsequently sensitivity to sample size (*n* steps) is relatively higher in lower pressure regions of the foot, and particularly the midfoot (Fig. 6B). As a result, many subjects show a range in mean values in excess of 25% of the full data set mean for the midfoot at less than 300 steps, and on average, our subjects yield a range of mean values above 100% of the full data set mean when less than 15 steps were used (Fig. 6B). This contrasts considerably with higher pressure regions in the heel and forefoot where the mean range for all subjects only reaches 10–30% the value recovered using all steps (n >500 steps) when less than 10 steps are used to calculate the mean (Fig. 6).

The range of whole-foot mean MSE at low sample sizes ($n < 10$) is 75% higher than the overall dataset mean MSE range of <500 (Fig. 7). Furthermore, the magnitude of the sensitivity of the mean MSE to subsample $n$, is highly variable between individuals (Fig. 7). For example, at a walking speed of 1.3 m/s and a subsample $n = 10$, subject 7 (highest MSE) and subject 13 (lowest MSE), had a MSE range of 10.3 and 2.2 respectively. However, at $n = 400$, their MSEs were reduced to 1.5 and 0.4 respectively (Fig. 7) and to within 5% of the overall dataset mean for 1.3 m/s. This result suggests that many pressure records ($n > 400$) are required to reflect the high levels of observed variation in peak pressure given by the full dataset (Fig. 7). This suggests that at small sample sizes ($n = <25$), there is a high probability that neither the range of variation nor the mean peak pressure will be reflected as accurately as it would if collected from larger datasets ($n > 400$) (Fig. 7).

Our finding that a relatively high number of steps are required to capture a high proportion of the variability in different pressure metrics present in the full datasets (e.g., $n > 100$ steps in mMEANP and mMAXP, Figs. 3–4; $n > 300$ steps for single 'midfoot' pixel values, Fig. 6E; $n > 400$ steps in whole-foot MSE values, Fig. 7), compliments other reliability studies that assess variability in kinematic data (e.g., Owings & Grabiner, 2003; Bruijn et al., 2013). However, these results are not entirely consistent, at least qualitatively, with previous reliability studies specific to plantar pressure (e.g., Kernozek, LaMott & Dancisak, 1996; Keijsers et al., 2009; Arts & Bus, 2011), and suggest that a considerably larger $n$ is necessary to capture pressure patterns. As mentioned previously, earlier clinical assessments of variability in plantar pressure concluded that between 4–12 pressure records to be a valid and reliable interpretation of peak plantar pressure in patients with diabetic neuropathy (Kernozek, LaMott & Dancisak, 1996; Keijsers et al., 2009; Arts & Bus, 2011). As a general guide to capturing pressure characteristics, these small step numbers are lower than our results from the simple pressure metrics suggest are appropriate in healthy subjects (Figs. 3–4). Our results are more similar to those of Melvin et al. (2014) who suggested that a minimum of 30 steps per foot should be collected to accurately represent a subject's mMAXP. Variation across studies in their goals or hypotheses, experimental protocols (see discussion above), equipment, subject characteristics and health status, and in the analytical methods used all combine to make it difficult to assess why our findings suggest, at least qualitatively, that a relatively large number of steps are required to accurately represent the central tendency of an individual's peak pressure magnitude and distribution (Figs. 3–7) compared to previous work (Kernozek, LaMott & Dancisak, 1996; Keijsers et al., 2009; Arts & Bus, 2011).

Furthermore, our study perhaps considers 'reliability' in a different context to most, if not all, previous studies. Previous studies explicitly or implicitly view reliability in its classical sense: as the ratio of between- vs. within-subject variability. When this ratio is high, the data are generally considered to be 'reliable'. From this perspective, a relatively small sample size (e.g., $n = 10$ steps or less) is sufficient for generating reliable conclusions because between-subject variability is usually very large compared to trial-to-trial or step-to-step variability. In contrast to this classical perspective, our paper considered 'reliability' in terms of longer-term within-subject behaviour, which we defined in terms of metric value convergence over a large number of trials. Although this perspective is defined in terms of

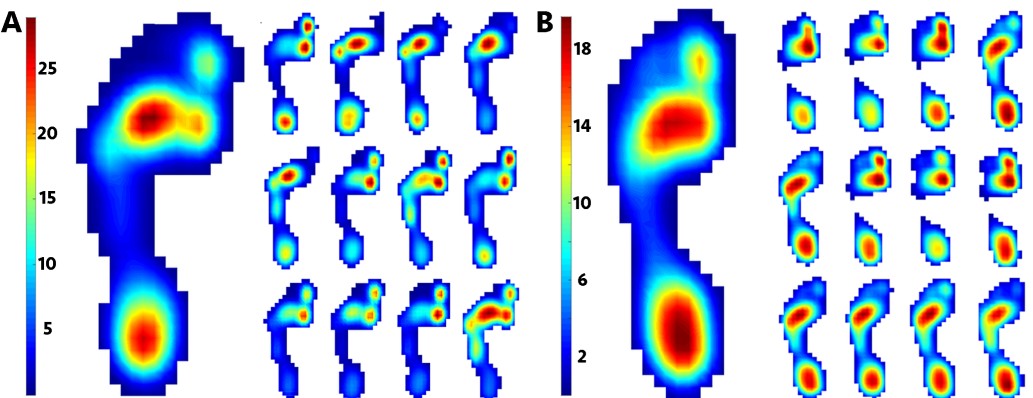

**Figure 8 Examples of intra-subject (step-to-step) variation in peak pressure magntidue and distribu-tion.** Mean peak plantar images (left, large images) and the peak plantar pressure records from the first 12 consecutive steps (right, smaller images) from the subjects with the highest (A, subject 7) and lowest (B, subject 13) MSE to illustrate high levels of step-to-step variability in pressure distribution at 1.3 m/s.

within-subject 'reliability', it also has important implications for between-subjects analysis, as demonstrated in Figure 4. The potential for miscalculating or misinterpreting pressure patterns because of small sample sizes can be further illustrated by visualizing the range in peak pressure in a small number of peak pressure records (Fig. 8). Figure 8 shows the first 12 steps at 1.3 m/s produced by the participants with the highest (subject 7) and lowest (subject 13) mean MSE in our pixel-level analysis (Fig. 7). The overall pressure distribution and the position of peak pressure shifts over the plantar surface in both participants step-to-step, and very few of these records qualitatively resemble the mean pressure image (Fig. 8). Furthermore, participants in this study did not show the large variance in midfoot pressure noted in other healthy participants in a previous study (*Bates et al., 2013a*) where midfoot pressure varied step-to-step from minimum to maximum pressure across the plantar surface in some individuals. Presumably, participants exhibiting such extreme ranges in mid-foot pressure would show even more variation over the plantar surface than those analysed herein.

Technological advances in data collection (e.g., instrumented treadmills) and analysis (e.g., automated image analysis algorithms and pixel-level statistical comparisons) are making interrogation of very large biomechanical datasets increasingly feasible. These advances are important given the revelation of seemingly high levels of intra-subject or step-to-step variance in pressure magnitude and distribution in recent analyses of larger than average pressure datasets (*Bates et al., 2013a*; *McClymont et al., 2016*). Our results demonstrate that at the largest sample sizes typically collected in plantar pressure analyses ($n = 20$–$50$ records) the range of mean MSE values given by the 1000 randomly generated subsamples is more than 50% higher than the mean MSE of the full dataset of <500 (Fig. 7). At $n < 10$ records, this increases to more than 75%, indicating a high probability that such a sample of records will not reflect either the range of variance or the mean pressure given by a larger dataset of $n = 500$ (Fig. 7). Sample-size related reliability is specific to experimental goals and hypotheses thus caution must be taken when evaluating individual studies in the

context of these results. However, the potential implications of these results (Figs. 3–8) are clearly non-trivial, given that most hypotheses regarding human foot development, change and function, are derived from sample sizes of 3-30 pressure records per subject (*Hughes et al., 1991*; *Hennig & Rosenbaum, 1991*; *Cavanagh et al., 1997*; *Burnfield et al., 2004*; *Segal et al., 2004*; *Warren, Maher & Higbie, 2004*; *Drerup, Szczepaniak & Wetz, 2008*), with only three examples of single participant trials comprising $n = 50$ records (*Rosenbaum et al., 1994*; *Taylor, Menz & Keenan, 2004*; *Melvin et al., 2014*).

## ACKNOWLEDGEMENTS

This work is dedicated to the colourful life of Russell Savage, who sadly passed away before this work was published. Without his dedication and scientific creativity, this work would not have been possible. We also thank the many volunteer participants for giving up their time to help create such a large dataset. Thanks to Dr. Steven Lang for guidance on statistical terminology for reliability thresholds.

### Funding

This work was funded by the Institute of Ageing and Chronic Disease, University of Liverpool, a National Environment Research Council grant (NE/H004246/1) to R.H.C., and a Leverhulme Trust grant (RPG-2017-296) to K.T.B. The funders had no role in study design, data collection and analysis, decision to publish, or preparation of the manuscript.

### Grant Disclosures

The following grant information was disclosed by the authors:
The Institute of Ageing and Chronic Disease, University of Liverpool, a National Environment Research Council grant: NE/H004246/1.
A Leverhulme Trust grant: RPG-2017-296.

### Competing Interests

The authors declare there are no competing interests.

### Author Contributions

- Juliet McClymont and Karl T. Bates conceived and designed the experiments, performed the experiments, analyzed the data, prepared figures and/or tables, authored or reviewed drafts of the paper, and approved the final draft.
- Russell Savage conceived and designed the experiments, performed the experiments, prepared figures and/or tables, and approved the final draft.
- Todd C. Pataky conceived and designed the experiments, performed the experiments, authored or reviewed drafts of the paper, and approved the final draft.
- Robin Crompton performed the experiments, authored or reviewed drafts of the paper, and approved the final draft.
- James Charles performed the experiments, analyzed the data, authored or reviewed drafts of the paper, and approved the final draft.

## Data Availability

We used a freely available human plantar pressure dataset previously published in *McClymont et al. (2016)* and available at Dryad: McClymont, J, Pataky, TC, Crompton, RH, Savage, R, Bates, KT (2016), Data from: The nature of functional variability in plantar pressure during a range of controlled walking speeds, Dryad, Dataset, https://doi.org/10.5061/dryad.09q2b.

## Supplemental Information

Supplemental information for this article can be found online at http://dx.doi.org/10.7717/peerj.11660#supplemental-information.

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
