# Peer review of "Intra-subject sample size effects in plantar pressure analyses"

_PeerJ, doi:10.7717/peerj.11660_

## Round 0.1 · original submission · Major Revisions

Several issues have been indicated by the reviewers which you should address in a revised version of the text, taking into account that all of them have highlighted scientific merit in your work.

·

Basic reporting

The manuscript is fairly clear, well organised, and self-contained. The plantar pressure data is publicly-available and easy to access.

Regarding the cited literature, it is generally good. However, I would also add the following paper to the references:

* Keijsers, N. L. W., Stolwijk, N. M., Nienhuis, B., & Duysens, J. (2009). A new method to normalize plantar pressure measurements for foot size and foot progression angle. Journal of Biomechanics, 42(1), 87–90. doi:10.1016/j.jbiomech.2008.09.038

Note Fig. 4 in that paper and the discussion thereof. It uses intra-class correlation to estimate sample sizes for plantar pressure studies and is therefore highly relevant.

Regarding manuscript clarity, there are some clarity issues in the data analysis section. Most notably, it is not clear when the authors intend to use what they call simple metrics, and when they use vector analysis. In particular, it is unclear when pSPM is used. Also, equation 1 has some formatting problems (I assume that the pixel number, k, should be in brackets and noted on the sum).

Experimental design

The research question is clear and meaningful. The design of the experiments is generally good. I am happy to see the authors using statistical bootstrapping techniques to evaluate different sample sizes. I believe that that is the right way go do. I am also happy to see the authors evaluating sample sizes at different walking speeds.

My one criticism is that the authors aggregate their results across the entire plantar surface in equation 1. In quantitative plantar pressure analyses, it is very rare to aggregate results over the whole plantar surface. Usually, results are presented per pixel or aggregated over regions of interest. By reporting a single result over the whole plantar surface, as is done in this manuscript, it becomes difficult to interpret what impact the sample sizes have on traditional plantar pressure analyses. For example, I would expect that a region-of-interest plantar pressure study might require less samples than a pixel-by-pixel study because the within-region variability gets discarded before the statistical analysis is performed. Also, the paper I mentioned earlier by Keijsers et al. suggests that different foot regions require different numbers of samples in order to get reliable estimates of the mean peak pressures. If my study were to focus only on, say, the hallux, would I still need 400+ samples? Given the way the experiments were setup in this paper, I cannot get a clear answer to this question.

I feel that the paper would be much stronger if the authors were to report their sample size results on a per region or per pixel level. This way, I can easily determine how many plantar pressure samples I should collect based on the foot region I’m interested in and the type of statistical analysis I intend to perform. The current results do not make that easy to figure out.

Validity of the findings

The results presented agree with what I would expect and their interpretation is performed properly. I am happy to see that the authors have listed the limitations of their study, particularly the use of the treadmill for measurement and the choice of pressure metric.

The one criticism I have is on the discussion of the sample sizes recommended in this study. The authors note in their discussion section that multiple studies have recommended between 8-30 samples being necessary for reliable mean plantar pressure estimates. The same can be said for the work of Keijsers et al. that I mentioned earlier. However, the authors propose that around 400 samples are required. This is a difference of an order of magnitude. It would be nice if the authors could comment on why there is such a large discrepancy between their results and those from previous studies. Is it because of the choice of pressure metric, the choice of analysis technique (e.g. region-of-interest, per pixel), the use of the treadmill, the reliability of the sensors, or something else? Some extra discussion here would be very helpful.

Additional comments

Overall, I am very enthusiastic about the paper and would like to see it get published. That being said, I feel that the paper can be made much stronger by (a) reporting results at either the per region or per pixel level, and (b) further discussing the sample size discrepancy between their results and those in previous studies.

Reviewer 2 ·

Basic reporting

This paper considers within-participant variability and the effect that the recording of multiple steps has on plantar pressure metrics. Overall this is a relevant and scientifically valid paper that addresses an important issue across a number of fields of study where these measurements are used. The authors consider both “traditional” and novel “pixel based” interpretations which provides additional rigour and depth to the results. I generally support the approach to evaluating the data that the authors took and found the results both novel and interesting. I did however find the manuscript results in the text a little difficult to follow in places and the figures could be improved to showcase the results much more effectively. Below are a number of comments I would like the authors to consider.

Major comment 1.
The authors use the terms “sample size”, “steps”, “pressure records” to refer to the number of within-participant trials. I think that the term “sample size” is potentially confusing as this typically refers to the number of participants used within an experiment rather than the number of trials used to get a stable estimate of the within-participant variability. Sample size also is inherently linked to experimental power which is not directly considered in this manuscript. I would consider “steps” to be most appropriate and I would remove sample size throughout.

Major comment 3. Habitual variance: This term appears multiple times in the introduction and is ultimately what this paper tries to address however there is no definition of this provided. The authors comment on the effects on habitual variation in the discussion too. I therefore think that some definition of this and perhaps an explicit statement of how the authors intend to measure this (see comment #2 below) should be added to the introduction.

Major comment 4. Figures: Unfortunately I did not find the figures to be very information or clear to understand. Some attention should be given to making these more clear both visually and in the descriptions.

Figure 1: The markers were too small and the grey data points were barely visible. The key data <50 steps was quite squashed. Why use different coloured markers and lines?
The x and y-axis labels are ambiguous- use simple terms e.g. Range of mean Max Pressure and number of steps. The term “N records in randomly generated subsample” is confusing as you could misinterpreted this as 1000 records were used for each.
Personally I would have liked to have seen the distributions created for each subsampling visualised and for the breadth of the distribution calculated. The wider the distribution, the greater uncertainty around the mean.

Figure 2: This is confusing because you manipulate both time (top row) and subjects (bottom row) without clearly describing this. Furthermore you refer to the “range” yet this is not used in the y-axis label.

Figure 3: The legend should be much more concise. The data in the smaller samples are unclear. The x axis label is different in this compared to the other figures.

Figure 5: Excellent figure. The colourbar labels are not readable.

Experimental design

Velocity units are incorrect in figure captions but check throughout. Should be either m/s or m.s-1.

The authors have conducted a rigourous investigation.

Validity of the findings

Major Comment 2.
For the pixel level analysis the authors provide a quantitative indicator of the “stability” the results – the number of records required for the range in mean MSE to be within 5% of the overall dataset mean. The aim of the authors is to understand the “stability” of the results and therefore I think this or a similar approach should be applied to the singular pressure metrics described in figure 1. Including this will allow the authors to describe the results more precisely. Currently it is much easier to consider the consequences of the pixel level approach than it is the singular metrics because of statements like “MSE only came within 5% of the overall sample size in subsamples >400” versus “the range of basic metric values….increased considerably”. Once this is included I would also considering a table of results as currently it is difficult to extract the key results from the figures except for figure 4.


With the above results it would then be nice to add more specifically to the end of the abstract the number of steps required for the measures to become stable for all approaches taken.

Additional comments

Minor comments

L78 Kinematic parameter OF GAIT suggest multiple steps

L96 How is the reliability of these results being determined – are you planning to use the same approach?

L148 Unfortunately I wouldn’t consider 2008 “recently” anymore….

L150 Please clarify whether this should be deviations from length and width in both part of the sentence or why length only for the registration.

L156 rephrase – foot pressure variance to sample size

L169 Please clarify was this the peak of the whole foot and mean of the whole foot?

L170/1 – This description (“from all five speeds”) seems to indicate that the MSE was calculated across speeds but I don’t think this is the case from the equation – please clarify.

L182 – Trial

L192 – I would reverse this sentence by saying to only present the 1.3 m/s data and the others are available elsewhere.

L255 – This first sentence is long and unclear – please rephrase.

L257 – “the effects of n subsamples….provide general guidance” – As mentioned I don’t believe these results do this easily. Consider main comment #2 and a summary table.

L265 – “This is so…..” – phrasing is awkward.

L280 – Could the phrase “comparing treadmill to overground” be clearer than “quantitative results to non-consecutive datasets”?

L287 – Your study does not consider other hypotheses or research questions therefore I would remove this part of the sentence.

L319 – this is an example of where some more precise stability results would allow you to make more precise statements.

---

## Round 0.2 · accepted · Accept

All the reviewers' concerns have been correctly addressed.

·

Basic reporting

The revised manuscript is clear and complete.

Experimental design

The experiments are thorough and have been performed well. I am happy to see the inclusion of the regional results and I find them particularly interesting. I thank the authors for including them.

Validity of the findings

The results presented agree with what I expected and the conclusions logically follow from the results presented. I was particularly interested to see that the regional within-subject variances converged more slowly than the whole foot measurements. I was also happy to get the authors' feedback on why their plantar pressure "reliability" results differed from previously-reported studies.

Additional comments

I once again thank the authors for their thorough revision of the manuscript. I feel that this work would be an excellent addition to the field.

Reviewer 2 ·

Basic reporting

I am satisfied with the changes and responses that the authors have made in this section based on my first review.

Experimental design

I am satisfied with the changes and responses that the authors have made in this section based on my first review.

Validity of the findings

I am satisfied with the changes and responses that the authors have made in this section based on my first review.

Additional comments

I would like to thank the authors for responding to my comments in a thorough and comprehensive manner. I am satisfied with the changes and responses that the authors have made based on my first review.